# Parental compliance and reasons for COVID-19 Vaccination among American children

**Neil K. R. Sehgal**[1,2]*, **Benjamin Rader**[1,3], **Autumn Gertz**[1], **Christina M. Astley**[1,4,5], **John S. Brownstein**[1,4]

**1** Boston Children's Hospital, Boston, Massachusetts, United States of America, **2** Institute for Applied Computational Science, Harvard University, Cambridge, Massachusetts, United States of America, **3** Boston University School of Public Health, Boston, Massachusetts, United States of America, **4** Harvard Medical School, Boston, Massachusetts, United States of America, **5** Broad Institute of MIT and Harvard, Cambridge, Massachusetts, United States of America

* neil_sehgal@g.harvard.edu

**Data Availability Statement:** Data and code used for analyses, tables, and figures are available here:

## Abstract

COVID-19 vaccination rates among children have stalled, while new coronavirus strains continue to emerge. To improve child vaccination rates, policymakers must better understand parental preferences and reasons for COVID-19 vaccination among their children. Cross-sectional surveys were administered online to 30,174 US parents with at least one child of COVID-19 vaccine eligible age (5–17 years) between January 1 and May 9, 2022. Participants self-reported willingness to vaccinate their child and reasons for refusal, and answered additional questions about demographics, pandemic related behavior, and vaccination status. Willingness to vaccinate a child for COVID-19 was strongly associated with parental vaccination status (multivariate odds ratio 97.9, 95% confidence interval 86.9–111.0). The majority of fully vaccinated (86%) and unvaccinated (84%) parents reported concordant vaccination preferences for their eligible child. Age and education had differing relationships by vaccination status, with higher age and education positively associated with willingness among vaccinated parents. Among all parents unwilling to vaccinate their children, the two most frequently reported reasons were possible side effects (47%) and that vaccines are too new (44%). Unvaccinated parents were much more likely to list a lack of trust in government (41% to 21%, p < .001) and a lack of trust in scientists (34% to 19%, p < .001) as reasons for refusal. Cluster analysis identified three groups of unwilling parents based on their reasons for refusal to vaccinate, with distinct concerns that may be obscured when analyzed in aggregate. Factors associated with willingness to vaccinate children and reasons for refusal may inform targeted approaches to increase vaccination.

## Author summary

Despite wide availability of vaccines in the US, many children remain unvaccinated against COVID-19. We sought to understand why some parents in the US are unwilling to vaccinate their children against COVID-19. To do this, an online survey was conducted of 30,174 parents with children aged 5–17. We found that parents who were vaccinated

https://github.com/sehgal-neil/parental-vaccine-hesitancy.

**Funding:** This work was supported by the National Institute of Diabetes and Digestive and Kidney Diseases of the National Institutes of Health (K23 DK120899 to CMA) and the Centers for Disease Control and Prevention (Contract Number 75D30121C11606 to BR, AG and JB). BR, AG & JB were funded on the same contract. The funders had no role in study design, data collection and analysis, decision to publish, or preparation of the manuscript.

**Competing interests:** The authors have declared that no competing interests exist.

themselves were more likely to vaccinate their children. Parents who were unwilling to vaccinate their children cited side effects and the newness of the vaccines as the most common reasons. However, the majority of unwilling parents did not cite side effects as a reason for their unwillingness, suggesting a wide variety of concerns across parents. Moreover, among unwilling parents, unvaccinated parents were more likely than vaccinated parents to cite a lack of trust in government or scientists. The findings suggest that targeted approaches are needed to increase vaccination, taking into account the specific concerns of different groups of parents.

## Introduction

In the United States (US), children as young as 5 years old became eligible for the Pfizer-BioNTech COVID-19 vaccine in October 2021 [1]. Studies of the vaccine in real world settings have found it to be highly effective in preventing severe disease and reducing hospitalization among children and adolescents [2,3]. However, despite wide availability of vaccines in the US, many children remain unvaccinated and vaccine uptake appears to have slowed [4]. On May 25, 2022, just 51% of children ages 5–17 had received at least one dose of a COVID-19 vaccine. By September 21, 2022, rates had risen by just one percentage point to 52% [5]. This is a lower rate compared to coverage levels of many other similarly recommended immunizations seen in the past. For instance, in the 2018–2019 school year, 63% of children between 6 months and 18 years received a flu vaccine [6]. Likewise, the coverage rate for the meningococcal conjugate (MenACWY), a disease with similarly rare but severe effects, was 89% among adolescents in 2020 [7].

According to the Centers for Disease Control and Prevention (CDC), as of September 2022, there have been over 15 million COVID-19 cases, over 12,200 laboratory-confirmed COVID-19 associated hospitalizations, and over 1,700 COVID-19 deaths among children in the United States (US) [8,9]. While adults have faced much higher acute morbidity and mortality, long term impacts of the disease on children are still unclear. Recent estimates from the United Kingdom suggest that around 5% of all secondary school students have experienced Post-COVID Conditions [10]. Additionally, children have faced far-reaching secondary effects from the pandemic, including disrupted healthcare services, routine immunizations, and education, and an increased prevalence of mental health disorders such as anxiety and depression [11].

Given the large disparities in vaccine coverage between COVID-19 and routine pediatric immunizations, increasing COVID-19 vaccination rates among children may require new strategies. Much prior work has been devoted to understanding COVID-19 compliance among parents. Prior studies have found vaccinated parents much more likely to vaccinate their children and that side effects and vaccine safety are two of the most common reasons for concern [4,12–22]. However, prior studies often analyzed all parents as a single cohort, rather than investigating this heterogeneity. Additionally, prior research has been limited by potentially non-representative samples. In this large, diverse, and nationally representative study, we aim to identify the demographic factors associated with parental compliance for COVID-19 vaccination among children in the United States and reasons for refusal. We also examine heterogeneity among vaccinated and unvaccinated parents through the inclusion of numerous relevant predictor variables and detailed information on reasons for refusal.

## Methods

To understand compliance for child vaccination against COVID-19, we administered an online survey focusing on parents with children ages 5–17, and analyzed responses using survey weights, logistic regression, and cluster analysis.

### Participants

Each day, over 2 million individuals across the world complete online surveys using the SurveyMonkey platform by Momentive [23]. Between January 1 and May 9, 2022, a random subset of US SurveyMonkey participants were invited to complete an additional anonymous questionnaire on COVID-19. There were no financial incentives for completing the survey, with 217,023 of those invited choosing to participate. Primary analysis restricted to parents ages 18+ who self-reported eligible children ages 5 to 17 (i.e. excluded parents with any ineligible children <5 yr). To evaluate the impact of excluding parents of children < 5 years, which could be a proxy for family size, we conducted a sensitivity analysis without this exclusion (S1 Table).

### Survey questions

Questions were selected from a previously validated web-survey [24,25]. Participants answered questions about demographics, pandemic related behavior, and vaccination status. Participants who received at least one dose of the vaccine were classified as vaccinated. Participants were asked if they had children of the following ages: under 5, 5 to 11, 12 to 15, and 16 to 17. For each age group that parents had children of, they were additionally asked if they were willing to have their child(ren) of that age group vaccinated against COVID-19. Participants could respond with"yes", "no", or "not sure". To understand parents with clear preferences we restricted our analysis to parents always willing or unwilling to vaccinate all of their children. Participants who were unwilling to vaccinate at least one of their children were asked to select up to 20 non-exclusive l reasons for their unwillingness from a list (S2 Table).

### Analyses

Using the latest estimates from the U.S. Census Bureau's American Community Survey, survey weights were applied to reflect the US adult population in demographic composition and political beliefs [26]. Census derived weights accounted for age, race/ethnicity, sex, education, and geography. To account for political beliefs, an additional smoothing parameter for political party identification was included based on aggregates of SurveyMonkey research surveys [24]. Weights were generated from all 217,023 respondents. Weights totaled 0.98 because the sample that met the inclusion criteria were not fully representative of the overall study population, with higher weighted individuals marginally more likely to be excluded. A previous analysis comparing the weighted and unweighted populations found weighted populations closely approximated the demographic composition of US adults [25].

Weighted multivariate logistic regression models were used to assess the association between parental characteristics and willingness to vaccinate their children. The estimation equation is:

$$log\left(P(y=1)/P(y=0)\right) = \alpha + \beta X_i + Z_i^{'}\gamma + \epsilon_i \qquad (1)$$

where $y$ is a binary variable for the willingness of parent i to vaccinate all of their children, $\beta$ represents the increase in log odds of parent i's willingness to vaccinate their child associated with parental vaccination status, $X_i$ is a categorical variable representing four levels of parental

vaccination status with unvaccinated as the reference level, $Z_i$ is a vector of parent level factors, $\gamma$ represents the increase in log odds of parent i's willingness to vaccinate their child associated with each parent level factor, and $\epsilon_i$ represents the error term. The estimate equation assumes that $\beta$ is constant across any parent level differences. Observations are weighted by their census derived survey weights.

For robustness, we conduct a stratified analysis to verify that the effect of parental vaccination status on willingness is consistent across levels of predictors. In order to understand the representativeness of our sample and generalizability of results, we additionally conduct an 80/20 test train split validation. Because the set of parent level factors that increase the likelihood of being vaccinated in parents may also increase the likelihood of being willing to vaccinate children, we additionally model parental willingness without parental vaccination status.

To assess heterogeneity in correlates of willingness between vaccinated and unvaccinated parents, we fit regressions separately for fully vaccinated and unvaccinated parents, as well as for the overall sample. We used KModes cluster analysis, an extension of KMeans for categorical data, to group unwilling parents into clusters according to their reasons for reluctance [27]. All statistical analyses were conducted in R version 4.1.2. Analyses used the cluster version 2.1.2, survey version 4.1.1, and gtsummary version 1.6.0.9008 R packages [28–30]. This study was approved by the Boston Children's Hospital Institutional Review Board (IRB-P00023700) and received a waiver of informed consent. No individuals under 18 were enrolled in the study.

## Results

Of the 217,023 survey respondents, 3,301 were under the age of 18 and excluded. An additional 159,752 users were excluded for not having children 17 or under and 10,098 users were excluded for having children under the age of 5. Next, 9271 respondents were excluded for missing data on variables of interest. Lastly, 3,410 parents were removed who were always unsure about vaccinating their children, and 1,051 were removed for only intending to vaccinate some of their children, leaving a total sample of 30,140. After exclusion, the mean survey weight of participants was .98, leading to a weighted sample of 29,583 (S1 Fig).

Of the 29,583 surveyed parents, 7,654 (26%) were unvaccinated, 2,697 (9%) were partially vaccinated, 9,713 (33%) were fully vaccinated, and 9,519 (32%) were fully vaccinated and boosted. A total of 19,586 (66%) parents expressed willingness to vaccinate their child. Parent characteristics are listed in Table 1. About half of participants were female, and a majority were White, religious, employed, and insured. A plurality of parents were ages 40–49 years, and had a household income under $49,999. S3 Table displays summary characteristics of parents who were always unsure or held inconsistent preferences across their children.

We observed a high degree of concordance in preferences, with 86% of vaccinated parents willing to have their children vaccinated and 85% of unvaccinated parents unwilling to have their children vaccinated. We observe that 70% of partially vaccinated parents, 77% of fully vaccinated parents, and 95% of fully vaccinated and boosted parents are willing to have their children vaccinated (S4 Table). In a pooled regression of all parents, willingness to vaccinate children was independently associated with a number of factors (Table 2). Major predictors of parents' preferences for child vaccination included being fully vaccinated and boosted (odds ratio [OR]: 106.0, 95% confidence interval [CI]: 93.9–120.0), Democrat political affiliation (OR: 4.22, CI: 3.82–4.66), and Asian race (OR: 3.07, CI: 2.56–3.71) (Fig 1). We find statistical differences between all parental vaccination status levels regardless of reference level chosen (S5 Table). VIF analysis suggested there exist limited cross-correlation among correlates (squared scaled GVIF threshold of 5). A sensitivity test across age, education, income, and

**Table 1. Characteristics of studied participants.**

| | Weighted (N = 29,583) | Unweighted (N = 30,140) |
|---|---|---|
| | n (%) | n (%) |
| Gender | | |
| Female | 15,304 (52%) | 18,915 (63%) |
| Male | 13,949 (47%) | 10,885 (36%) |
| Transgender or Nonbinary | 330 (1%) | 340 (1%) |
| Age | | |
| 18–29 years | 1,686 (6%) | 1,050 (3%) |
| 30–39 years | 8,018 (27%) | 6,616 (22%) |
| 40–49 years | 12,090 (41%) | 13,206 (44%) |
| 50–64 years | 6,872 (23%) | 8,473 (28%) |
| 65+ years | 917 (3%) | 795 (3%) |
| Household Income | | |
| Under $49,999 | 11,054 (37%) | 9,137 (30%) |
| $50,000-$99,999 | 8,384 (28%) | 8,237 (27%) |
| Over $100,000 | 10,146 (34%) | 12,766 (42%) |
| Race/Ethnicity | | |
| White, not Hispanic | 17,150 (58%) | 17,609 (58%) |
| Hispanic | 5,947 (20%) | 4,859 (16%) |
| Black | 5,947 (20%) | 4,859 (16%) |
| Asian | 1,677 (6%) | 1,621 (5%) |
| Other | 1,169 (4%) | 1,346 (4%) |
| Education | | |
| High School or Less | 11,294 (38%) | 5,762 (19%) |
| Some College | 9,075 (31%) | 8,894 (30%) |
| College Graduate | 9,215 (31%) | 15,484 (51%) |
| Employment Status | | |
| Employed | 24,388 (82%) | 25,449 (84%) |
| Unemployed | 5,195 (18%) | 4,691 (16%) |
| Health Insurance | | |
| Insured | 27,253 (92%) | 28,403 (94%) |
| Uninsured | 2,331 (8%) | 1,737 (6%) |
| Self Reported Health | | |
| Fair/Poor | 2,388 (8%) | 2,274 (8%) |
| Good | 7,588 (26%) | 7,598 (25%) |
| Very good | 11,194 (38%) | 11,771 (39%) |
| Excellent | 8,413 (28%) | 8,497 (28%) |
| Religious Status | | |
| Religious | 22,484 (76%) | 22,536 (75%) |
| Atheist/Agnostic | 7,099 (24%) | 7,604 (25%) |
| Parent COVID-19 Vaccination | | |
| Johnson | 1,970 (7%) | 1,856 (6%) |
| mRNA | 19,959 (67%) | 22,213 (74%) |
| Unvaccinated | 7,654 (26%) | 6,071 (20%) |
| Have Child Age 5 to 11 Years | 15,357 (52%) | 14,679 (49%) |
| Have Child Age 12 to 15 Years | 14,188 (48%) | 14,783 (49%) |
| Have Child Age 16 to 17 Years | 9,828 (33%) | 10,667 (35%) |
| Political Party Affiliation | | |

*(Continued)*

**Table 1.** (Continued)

|  | Weighted (N = 29,583) | Unweighted (N = 30,140) |
|---|---|---|
|  | n (%) | n (%) |
| Republican | 9,712 (33%) | 7,504 (25%) |
| Democrat | 9,538 (32%) | 11,402 (38%) |
| Independent | 10,334 (35%) | 11,234 (37%) |
| When Will the Pandemic End? |  |  |
| Already Over | 6,541 (22%) | 5,615 (19%) |
| Less than three months | 1,662 (6%) | 1,585 (5%) |
| Between three months and one year | 5,526 (19%) | 6,070 (20%) |
| More than one year | 15,854 (54%) | 16,870 (56%) |
| Flu Vaccine Since June 2021 | 13,935 (47%) | 15,924 (53%) |

political party finds the effect of parental vaccination status on willingness is consistent across parent level subgroups (S6, S7, S8 and S9 Tables). The model achieved an accuracy of 0.87 using a 50% cut off probability and an AUC of 0.91 when training on 80% of the sample and testing on the remaining 20%. In a model not using parent vaccination status as a determinant of willingness, we found that major predictors of parents' preferences for child vaccination still include Democrat political affiliation (OR: 6.03, CI: 5.57–6.53) and Asian race (OR: 5.15, CI: 4.37–6.09). In addition, we found that high income and having health insurance were independently positively associated with willingness, whereas we observed negative associations in the model including parental vaccination status.

Several correlates of willingness differed in our separated regression models of unvaccinated and fully vaccinated parents (Table 2). Among fully vaccinated parents, willingness was independently positively associated with older age, college education, atheism/agnosticism, receiving an mRNA vaccine, and no vaccine side effect of headache, nausea/vomiting, or rash. Among unvaccinated parents, willingness was independently positively associated with male gender, younger age, high school or less education, and a lack of health insurance.

The 9,998 parents who were unwilling to vaccinate their children listed several reasons for refusal (S2 Table). Overall, parents were most concerned about potential side effects (47%) and the vaccine being too new (44%). Vaccinated and unvaccinated parents differed on several concerns. For example, unvaccinated parents were much more likely to list a lack of trust in government (41% to 21%, p < .001) and a lack of trust in scientists (34% to 19%, p < .001) as reasons for refusal. Parents with older children are more likely to say the decision to vaccinate should be the child's decision when they are of appropriate age while parents with younger children are more likely to say their children are too young to be vaccinated (S10 Table).

Cluster analysis of unwilling parents' reasons indicated that 3 clusters achieved a good balance of interpretability and cluster quality. These clusters differ significantly on both reasons for reluctance and parent characteristics (Table 3). For instance, Cluster 1 parents (32%) had a median of 8.0 reasons for refusal, were very likely to lack trust in government and scientists, were very likely to think the vaccine was too new and worry about potential side effects, and were likely to view the pandemic as already being over. Cluster 2 parents make up the majority of all unwilling parents (56%). Parents in this cluster had a median of 1.0 reasons for refusal and a majority (53%) had a high school or less education. Cluster 3 parents were the smallest group, had a median of 4.0 reasons for refusal, and were very likely to think the vaccine was too new and worry about potential side effects.

**Table 2. Multivariate relationship between parent characteristics and willingness to vaccinate children.**

| | Willingness to Vaccinate Children | | | |
|---|---|---|---|---|
| | **All Parents, Modeling Parental Vaccination Status†** | **All Parents, Not Modeling Parental Vaccination Status†** | **Unvaccinated Parents** | **Fully Vaccinated Parents‡** |
| | **Odds Ratio (95% confidence interval)** | **Odds Ratio (95% confidence interval)** | **Odds Ratio (95% confidence interval)** | **Odds Ratio (95% confidence interval)** |
| Gender | | | | |
| Female | — | — | — | — |
| Male | 1.16 (1.08, 1.25)*** | 0.98 (0.93, 1.04) | 1.43 (1.24, 1.65)*** | 0.96 (0.87, 1.05) |
| Transgender or Nonbinary | 0.71 (0.53, 0.97)* | 0.51 (0.40, 0.66)*** | 1.11 (0.67, 1.76) | 0.65 (0.43, 0.99)* |
| Age | | | | |
| 18–29 years | — | — | — | — |
| 30–39 years | 0.98 (0.85, 1.14) | 1.15 (1.02, 1.29)* | 0.62 (0.50, 0.78)*** | 1.13 (0.92, 1.37) |
| 40–49 years | 1.30 (1.12, 1.51)*** | 1.89 (1.68, 2.13)*** | 0.80 (0.64, 1.01) | 1.72 (1.40, 2.10)*** |
| 50–64 years | 1.50 (1.27, 1.76)*** | 2.69 (2.36, 3.07)*** | 0.78 (0.58, 1.05) | 2.21 (1.77, 2.76)*** |
| 65+ years | 1.30 (0.99, 1.70) | 3.93 (3.17, 4.88)*** | 0.84 (0.45, 1.51) | 2.14 (1.55, 2.99)*** |
| Household Income | | | | |
| Under $49,999 | — | — | — | — |
| $50,000-$99,999 | 0.75 (0.69, 0.83)*** | 0.94 (0.87, 1.01) | 0.64 (0.53, 0.75)*** | 0.84 (0.74, 0.95)** |
| Over $100,000 | 0.72 (0.65, 0.79)*** | 1.15 (1.07, 1.25)*** | 0.44 (0.35, 0.56)*** | 0.98 (0.86, 1.12) |
| Race/Ethnicity | | | | |
| White, not Hispanic | — | — | — | — |
| Hispanic | 1.84 (1.68, 2.02)*** | 2.11 (1.96, 2.27)*** | 1.82 (1.53, 2.16)*** | 1.78 (1.57, 2.02)*** |
| Black | 1.61 (1.43, 1.81)*** | 1.34 (1.22, 1.48)*** | 1.66 (1.36, 2.02)*** | 1.20 (1.02, 1.43)* |
| Asian | 3.07 (2.56, 3.71)*** | 5.15 (4.37, 6.09)*** | 1.94 (1.10, 3.30)* | 3.17 (2.54, 3.99)*** |
| Other | 1.12 (0.95, 1.33) | 1.09 (0.96, 1.25) | 1.07 (0.76, 1.49) | 1.09 (0.88, 1.37) |
| Education | | | | |
| High School or Less | — | — | — | — |
| Some College | 0.80 (0.73, 0.87)*** | 1.06 (0.99, 1.13) | 0.78 (0.67, 0.92)** | 0.91 (0.81, 1.01) |
| College Graduate | 0.97 (0.87, 1.07) | 1.65 (1.52, 1.78)*** | 0.64 (0.49, 0.82)*** | 1.32 (1.16, 1.49)*** |
| Employment Status | | | | |
| Employed | — | — | — | — |
| Unemployed | 1.58 (1.43, 1.74)*** | 1.08 (1.00, 1.16)* | 1.58 (1.36, 1.83)*** | 1.56 (1.35, 1.80)*** |
| Health Insurance | | | | |
| Insured | — | — | — | — |
| Uninsured | 1.23 (1.09, 1.40)** | 0.76 (0.69, 0.83)*** | 1.22 (1.02, 1.45)* | 1.14 (0.93, 1.41) |
| Self Reported Health | | | | |
| Fair/Poor | — | — | — | — |
| Good | 1.05 (0.91, 1.20) | 1.03 (0.92, 1.14) | 0.89 (0.70, 1.13) | 1.12 (0.93, 1.34) |
| Very good | 1.05 (0.92, 1.20) | 1.01 (0.91, 1.12) | 0.80 (0.64, 1.02) | 1.12 (0.94, 1.34) |
| Excellent | 0.99 (0.86, 1.13) | 0.84 (0.75, 0.93)** | 0.79 (0.62, 1.00)* | 0.97 (0.81, 1.16) |
| Religious Status | | | | |
| Religious | — | — | — | — |
| Atheist/Agnostic | 1.35 (1.24, 1.47)*** | 1.34 (1.25, 1.43)*** | 1.12 (0.95, 1.30) | 1.55 (1.39, 1.74)*** |
| Have Child Age 5 to 11 Years | | | | |
| No | — | — | — | — |
| Yes | 0.51 (0.47, 0.56)*** | 0.59 (0.55, 0.63)*** | 0.55 (0.47, 0.66)*** | 0.53 (0.47, 0.59)*** |
| Have Child Age 12 to 15 Years | | | | |

*(Continued)*

**Table 2.** (Continued)

| | Willingness to Vaccinate Children | | | |
|---|---|---|---|---|
| | **All Parents, Modeling Parental Vaccination Status†** | **All Parents, Not Modeling Parental Vaccination Status†** | **Unvaccinated Parents** | **Fully Vaccinated Parents‡** |
| | **Odds Ratio (95% confidence interval)** | **Odds Ratio (95% confidence interval)** | **Odds Ratio (95% confidence interval)** | **Odds Ratio (95% confidence interval)** |
| No | — | — | — | — |
| Yes | 1.05 (0.98, 1.14) | 0.92 (0.87, 0.98)** | 0.78 (0.67, 0.90)*** | 1.15 (1.04, 1.27)** |
| Have Child Age 16 to 17 Years | | | | |
| No | — | — | — | — |
| Yes | 1.34 (1.23, 1.46)*** | 1.04 (0.97, 1.11) | 1.21 (1.02, 1.43)* | 1.48 (1.32, 1.67)*** |
| Political Party Affiliation | | | | |
| Republican | — | — | — | — |
| Democrat | 4.22 (3.82, 4.66)*** | 6.03 (5.57, 6.53)*** | 3.76 (3.08, 4.59)*** | 5.33 (4.68, 6.07)*** |
| Independent | 1.55 (1.44, 1.68)*** | 1.56 (1.47, 1.66)*** | 1.55 (1.32, 1.83)*** | 1.72 (1.56, 1.91)*** |
| Parent Vaccination Status | | | | |
| Unvaccinated | — | | | |
| Partially Vaccinated | 11.9 (10.6, 13.3)*** | | | |
| Fully Vaccinated | 19.9 (18.2, 21.7)*** | | | |
| Fully Vaccinated and Boosted | 106 (93.9, 120)*** | | | |
| Parent Vaccination Type | | | | |
| Johnson | | | | — |
| mRNA | | | | 2.08 (1.82, 2.37)*** |
| Rash | | | | |
| Yes | | | | — |
| No | | | | 1.24 (1.05, 1.46)* |
| Nausea/Vomiting | | | | |
| Yes | | | | — |
| No | | | | 1.40 (1.19, 1.65)*** |
| Muscle Ache | | | | |
| Yes | | | | — |
| No | | | | 1.09 (0.97, 1.22) |
| Pain in Arm | | | | |
| Yes | | | | — |
| No | | | | 0.99 (0.90, 1.10) |
| Swelling in Arm | | | | |
| Yes | | | | — |
| No | | | | 1.09 (0.96, 1.24) |
| Fever | | | | |
| Yes | | | | — |
| No | | | | 1.09 (0.95, 1.25) |
| Chills | | | | |
| Yes | | | | — |
| No | | | | 0.99 (0.86, 1.13) |
| Tiredness | | | | |
| Yes | | | | — |

(*Continued*)

**Table 2.** (Continued)

| | Willingness to Vaccinate Children | | | |
|---|---|---|---|---|
| | **All Parents, Modeling Parental Vaccination Status†** | **All Parents, Not Modeling Parental Vaccination Status†** | **Unvaccinated Parents** | **Fully Vaccinated Parents‡** |
| | **Odds Ratio (95% confidence interval)** | **Odds Ratio (95% confidence interval)** | **Odds Ratio (95% confidence interval)** | **Odds Ratio (95% confidence interval)** |
| No | | | | 1.11 (0.99, 1.23) |
| Headache | | | | |
| Yes | | | | — |
| No | | | | 1.18 (1.05, 1.32)** |
| Something Else | | | | |
| Yes | | | | — |
| No | | | | 1.56 (1.33, 1.82)*** |

*p < .05;

**p < .01;

***p < .001

†All Parents includes parents who are unvaccinated, partially vaccinated, fully vaccinated, and fully vaccinated and boosted

‡Fully vaccinated includes parents who are either fully vaccinated or fully vaccinated and boosted and excludes those partially vaccinated

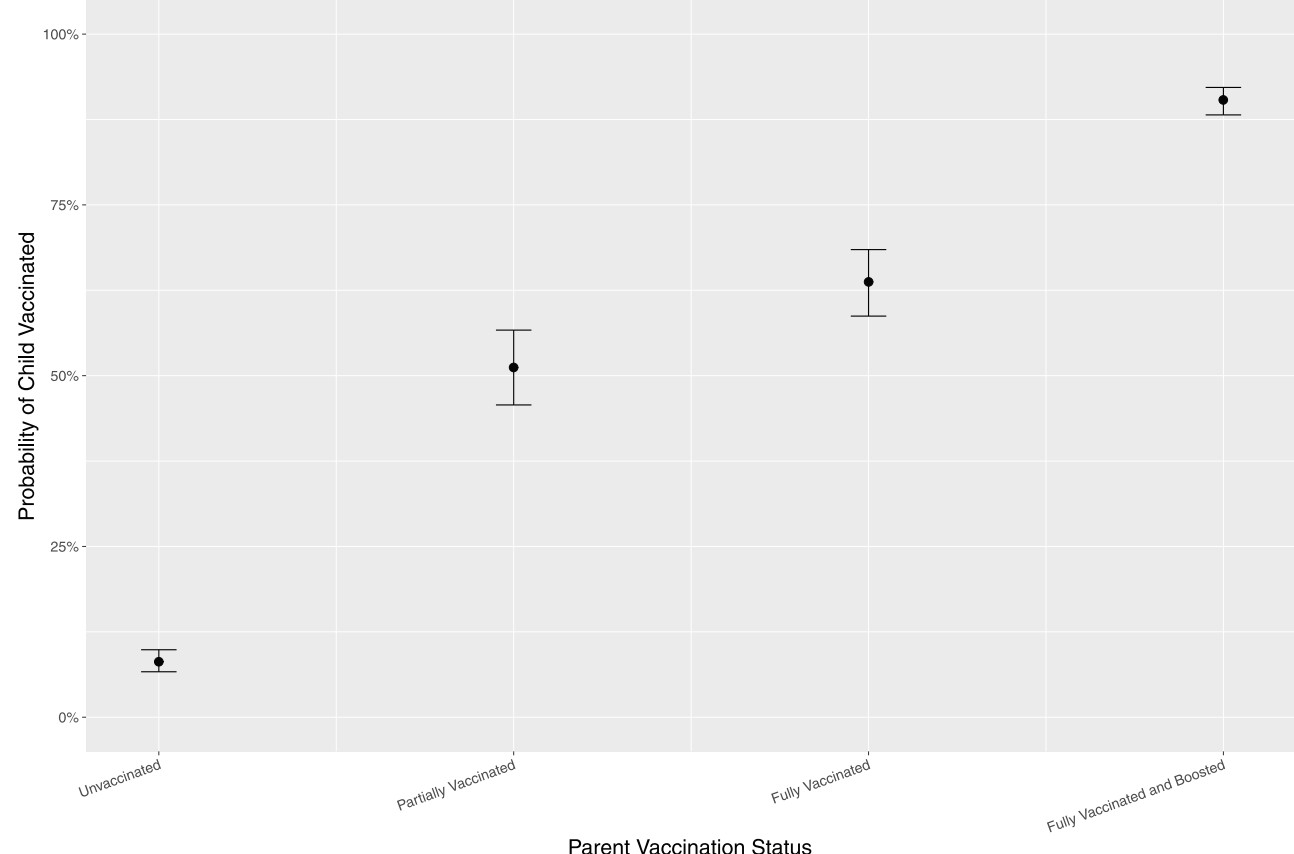

**Fig 1. Predicted probabilities of vaccinating child.**

**Table 3. Summary characteristics by clusters of unwilling parent.**

| | *All Unwilling Parents | Cluster 1 | | | Cluster 2 | | | Cluster 3 | | |
|---|---|---|---|---|---|---|---|---|---|---|
| | Overall, N = 9,998 | Unvaccinated, N = 2,342 | Vaccinated**, N = 847 | p-value† | Unvaccinated, N = 3,400 | Vaccinated**, N = 2,176 | p-value† | Unvaccinated, N = 733 | Vaccinated**, N = 500 | p-value† |
| **Characteristic** | | | | | | | | | | |
| Gender | | | | 0.004 | | | 0.28 | | | 0.88 |
| Female | 5,008/9,998 (50%) | 1,106/2,342 (47%) | 330/847 (39%) | | 1,685/3,400 (50%) | 1,124/2,176 (52%) | | 458/733 (62%) | 306/500 (61%) | |
| Male | 4,830/9,998 (48%) | 1,192/2,342 (51%) | 497/847 (59%) | | 1,662/3,400 (49%) | 1,030/2,176 (47%) | | 262/733 (36%) | 187/500 (37%) | |
| Transgender or Nonbinary | 159/9,998 (1.6%) | 44/2,342 (1.9%) | 20/847 (2.4%) | | 53/3,400 (1.6%) | 22/2,176 (1.0%) | | 13/733 (1.8%) | 7/500 (1.4%) | |
| Age | | | | 0.004 | | | <0.001 | | | 0.47 |
| 18–29 years | 859/9,998 (8.6%) | 119/2,342 (5.1%) | 35/847 (4.1%) | | 394/3,400 (12%) | 233/2,176 (11%) | | 54/733 (7.3%) | 25/500 (5.0%) | |
| 30–39 years | 3,732/9,998 (37%) | 847/2,342 (36%) | 234/847 (28%) | | 1,484/3,400 (44%) | 717/2,176 (33%) | | 263/733 (36%) | 187/500 (38%) | |
| 40–49 years | 3,757/9,998 (38%) | 930/2,342 (40%) | 367/847 (43%) | | 1,116/3,400 (33%) | 844/2,176 (39%) | | 306/733 (42%) | 194/500 (39%) | |
| 50–64 years | 1,492/9,998 (15%) | 408/2,342 (17%) | 186/847 (22%) | | 371/3,400 (11%) | 330/2,176 (15%) | | 107/733 (15%) | 89/500 (18%) | |
| 65+ years | 158/9,998 (1.6%) | 39/2,342 (1.7%) | 25/847 (3.0%) | | 35/3,400 (1.0%) | 52/2,176 (2.4%) | | 4/733 (0.5%) | 4/500 (0.7%) | |
| Household Income | | | | <0.001 | | | <0.001 | | | 0.092 |
| Under $49,999 | 4,135/9,998 (41%) | 798/2,342 (34%) | 182/847 (22%) | | 1,865/3,400 (55%) | 862/2,176 (40%) | | 273/733 (37%) | 153/500 (31%) | |
| $50,00-$99,999 | 3,026/9,998 (30%) | 733/2,342 (31%) | 262/847 (31%) | | 963/3,400 (28%) | 664/2,176 (31%) | | 242/733 (33%) | 162/500 (33%) | |
| Over $100,000 | 2,837/9,998 (28%) | 810/2,342 (35%) | 403/847 (48%) | | 572/3,400 (17%) | 650/2,176 (30%) | | 218/733 (30%) | 184/500 (37%) | |
| Race/Ethnicity | | | | 0.1 | | | <0.001 | | | 0.6 |
| White, not Hispanic | 6,839/9,998 (68%) | 1,834/2,342 (78%) | 653/847 (77%) | | 2,147/3,400 (63%) | 1,301/2,176 (60%) | | 553/733 (75%) | 350/500 (70%) | |
| Hispanic | 1,572/9,998 (16%) | 265/2,342 (11%) | 98/847 (12%) | | 586/3,400 (17%) | 444/2,176 (20%) | | 98/733 (13%) | 81/500 (16%) | |
| Black | 958/9,998 (9.6%) | 81/2,342 (3.5%) | 24/847 (2.9%) | | 507/3,400 (15%) | 270/2,176 (12%) | | 42/733 (5.7%) | 34/500 (6.8%) | |
| Asian | 183/9,998 (1.8%) | 31/2,342 (1.3%) | 27/847 (3.2%) | | 27/3,400 (0.8%) | 73/2,176 (3.3%) | | 13/733 (1.7%) | 13/500 (2.6%) | |
| Other | 445/9,998 (4.5%) | 130/2,342 (5.6%) | 44/847 (5.2%) | | 132/3,400 (3.9%) | 89/2,176 (4.1%) | | 27/733 (3.7%) | 22/500 (4.5%) | |
| Education | | | | <0.001 | | | <0.001 | | | <0.001 |
| High School or Less | 4,535/9,998 (45%) | 956/2,342 (41%) | 211/847 (25%) | | 2,009/3,400 (59%) | 953/2,176 (44%) | | 282/733 (38%) | 125/500 (25%) | |
| Some College | 3,333/9,998 (33%) | 838/2,342 (36%) | 335/847 (40%) | | 984/3,400 (29%) | 702/2,176 (32%) | | 285/733 (39%) | 188/500 (38%) | |
| College or More | 2,130/9,998 (21%) | 549/2,342 (23%) | 301/847 (36%) | | 406/3,400 (12%) | 521/2,176 (24%) | | 166/733 (23%) | 186/500 (37%) | |
| Employment Status | | | | <0.001 | | | <0.001 | | | 0.002 |
| Employed | 8,247/9,998 (82%) | 1,920/2,342 (82%) | 761/847 (90%) | | 2,625/3,400 (77%) | 1,896/2,176 (87%) | | 597/733 (81%) | 448/500 (90%) | |

(*Continued*)

**Table 3.** (*Continued*)

| | *All Unwilling Parents | Cluster 1 | | | Cluster 2 | | | Cluster 3 | | |
|---|---|---|---|---|---|---|---|---|---|---|
| | Overall, N = 9,998 | Unvaccinated, N = 2,342 | Vaccinated**, N = 847 | p-value† | Unvaccinated, N = 3,400 | Vaccinated**, N = 2,176 | p-value† | Unvaccinated, N = 733 | Vaccinated**, N = 500 | p-value† |
| Unemployed | 1,750/9,998 (18%) | 422/2,342 (18%) | 86/847 (10%) | | 775/3,400 (23%) | 280/2,176 (13%) | | 136/733 (19%) | 51/500 (10%) | |
| Health Insurance | | | | <0.001 | | | <0.001 | | | 0.058 |
| Insured | 8,917/9,998 (89%) | 2,103/2,342 (90%) | 815/847 (96%) | | 2,844/3,400 (84%) | 2,027/2,176 (93%) | | 657/733 (90%) | 472/500 (94%) | |
| Uninsured | 1,080/9,998 (11%) | 239/2,342 (10%) | 32/847 (3.7%) | | 556/3,400 (16%) | 150/2,176 (6.9%) | | 76/733 (10%) | 28/500 (5.6%) | |
| Self Reported Health | | | | 0.59 | | | 0.07 | | | 0.71 |
| Fair/Poor | 806/9,998 (8.1%) | 191/2,342 (8.1%) | 75/847 (8.9%) | | 273/3,400 (8.0%) | 172/2,176 (7.9%) | | 51/733 (7.0%) | 44/500 (8.7%) | |
| Good | 2,503/9,998 (25%) | 597/2,342 (26%) | 237/847 (28%) | | 811/3,400 (24%) | 497/2,176 (23%) | | 209/733 (28%) | 152/500 (30%) | |
| Very good | 3,628/9,998 (36%) | 881/2,342 (38%) | 315/847 (37%) | | 1,131/3,400 (33%) | 827/2,176 (38%) | | 287/733 (39%) | 188/500 (38%) | |
| Excellent | 3,061/9,998 (31%) | 673/2,342 (29%) | 220/847 (26%) | | 1,185/3,400 (35%) | 681/2,176 (31%) | | 186/733 (25%) | 116/500 (23%) | |
| Religious Status | | | | 0.45 | | | 0.002 | | | 0.14 |
| Religious | 7,870/9,998 (79%) | 1,812/2,342 (77%) | 670/847 (79%) | | 2,612/3,400 (77%) | 1,776/2,176 (82%) | | 581/733 (79%) | 419/500 (84%) | |
| Atheist/ Agnostic | 2,127/9,998 (21%) | 530/2,342 (23%) | 177/847 (21%) | | 787/3,400 (23%) | 400/2,176 (18%) | | 152/733 (21%) | 81/500 (16%) | |
| Have child age 5 to 11 | 6,214/9,998 (62%) | 1,347/2,342 (58%) | 549/847 (65%) | 0.005 | 2,084/3,400 (61%) | 1,451/2,176 (67%) | 0.003 | 436/733 (59%) | 348/500 (70%) | 0.005 |
| Have child age 12 to 15 | 4,725/9,998 (47%) | 1,257/2,342 (54%) | 405/847 (48%) | 0.03 | 1,586/3,400 (47%) | 858/2,176 (39%) | <0.001 | 396/733 (54%) | 223/500 (45%) | 0.013 |
| Have child age 16 to 17 | 2,792/9,998 (28%) | 784/2,342 (33%) | 228/847 (27%) | 0.009 | 962/3,400 (28%) | 481/2,176 (22%) | <0.001 | 237/733 (32%) | 100/500 (20%) | <0.001 |
| Party | | | | 0.051 | | | <0.001 | | | 0.021 |
| Republican | 4,767/9,998 (48%) | 1,236/2,342 (53%) | 487/847 (58%) | | 1,459/3,400 (43%) | 986/2,176 (45%) | | 339/733 (46%) | 260/500 (52%) | |
| Democrat | 1,231/9,998 (12%) | 102/2,342 (4.4%) | 48/847 (5.6%) | | 502/3,400 (15%) | 444/2,176 (20%) | | 69/733 (9.4%) | 67/500 (13%) | |
| Independent | 4,000/9,998 (40%) | 1,004/2,342 (43%) | 312/847 (37%) | | 1,439/3,400 (42%) | 747/2,176 (34%) | | 325/733 (44%) | 173/500 (35%) | |
| When will the Pandemic end? | | | | 0.16 | | | <0.001 | | | 0.002 |
| Already Over | 4,492/9,998 (45%) | 1,513/2,342 (65%) | 506/847 (60%) | | 1,344/3,400 (40%) | 653/2,176 (30%) | | 320/733 (44%) | 157/500 (31%) | |
| Less than three months | 560/9,998 (5.6%) | 66/2,342 (2.8%) | 38/847 (4.5%) | | 201/3,400 (5.9%) | 174/2,176 (8.0%) | | 45/733 (6.1%) | 35/500 (7.1%) | |
| Less than one year | 1,213/9,998 (12%) | 162/2,342 (6.9%) | 64/847 (7.6%) | | 495/3,400 (15%) | 323/2,176 (15%) | | 108/733 (15%) | 61/500 (12%) | |
| More than one year | 3,732/9,998 (37%) | 601/2,342 (26%) | 239/847 (28%) | | 1,359/3,400 (40%) | 1,027/2,176 (47%) | | 261/733 (36%) | 246/500 (49%) | |
| Flu vaccine since 6/21 | 2,111/9,998 (21%) | 189/2,342 (8.1%) | 314/847 (37%) | <0.001 | 345/3,400 (10%) | 971/2,176 (45%) | <0.001 | 68/733 (9.3%) | 224/500 (45%) | <0.001 |
| Parent Vaccination Status | | | | | | | | | | |

(*Continued*)

**Table 3.** (Continued)

| | *All Unwilling Parents | Cluster 1 | | | Cluster 2 | | | Cluster 3 | | |
|---|---|---|---|---|---|---|---|---|---|---|
| | Overall, N = 9,998 | Unvaccinated, N = 2,342 | Vaccinated**, N = 847 | p-value† | Unvaccinated, N = 3,400 | Vaccinated**, N = 2,176 | p-value† | Unvaccinated, N = 733 | Vaccinated**, N = 500 | p-value† |
| Unvaccinated | 6,475/9,998 (65%) | 2,342/2,342 (100%) | 0/847 (0%) | | 3,400/3,400 (100%) | 0/2,176 (0%) | | 733/733 (100%) | 0/500 (0%) | |
| Partially Vaccinated | 818/9,998 (8.2%) | 0/2,342 (0%) | 190/847 (22%) | | 0/3,400 (0%) | 529/2,176 (24%) | | 0/733 (0%) | 99/500 (20%) | |
| Fully Vaccinated | 2,223/9,998 (22%) | 0/2,342 (0%) | 569/847 (67%) | | 0/3,400 (0%) | 1,339/2,176 (62%) | | 0/733 (0%) | 315/500 (63%) | |
| Fully Vaccinated and Boosted | 482/9,998 (4.8%) | 0/2,342 (0%) | 88/847 (10%) | | 0/3,400 (0%) | 309/2,176 (14%) | | 0/733 (0%) | 85/500 (17%) | |
| Parent COVID-19 Vaccination Type | | | | | | | | | | |
| ohnson | | | 175/847 (21%) | | | 317/2,176 (15%) | | | 68/500 (14%) | |
| mRNA | | | 672/847 (79%) | | | 1,859/2,176 (85%) | | | 431/500 (86%) | |
| Local Adverse Effect | | | 679/847 (80%) | | | 1,539/2,176 (71%) | | | 417/500 (83%) | |
| Systemic Adverse Effect | | | 622/847 (73%) | | | 1,245/2,176 (57%) | | | 403/500 (81%) | |
| Rash | | | 120/847 (14%) | | | 182/2,176 (8.4%) | | | 60/500 (12%) | |
| Nausea/ Vomiting | | | 130/847 (15%) | | | 184/2,176 (8.5%) | | | 77/500 (15%) | |
| Muscle Ache | | | 439/847 (52%) | | | 709/2,176 (33%) | | | 299/500 (60%) | |
| Pain in Arm | | | 662/847 (78%) | | | 1,444/2,176 (66%) | | | 411/500 (82%) | |
| Swelling in Arm | | | 239/847 (28%) | | | 368/2,176 (17%) | | | 100/500 (20%) | |
| Fever | | | 266/847 (31%) | | | 497/2,176 (23%) | | | 183/500 (37%) | |
| Chills | | | 291/847 (34%) | | | 521/2,176 (24%) | | | 214/500 (43%) | |
| Tiredness | | | 497/847 (59%) | | | 891/2,176 (41%) | | | 325/500 (65%) | |
| Headache | | | 393/847 (46%) | | | 692/2,176 (32%) | | | 267/500 (53%) | |
| Something Else | | | 105/847 (12%) | | | 197/2,176 (9.1%) | | | 64/500 (13%) | |

†chi-squared test with Rao & Scott's second-order correction; Wilcoxon rank-sum test for complex survey samples

*All Parents includes parents who are unvaccinated, partially vaccinated, fully vaccinated, and fully vaccinated and boosted

**Vaccinated parents includes parents who are partially vaccinated, fully vaccinated, or fully vaccinated and boosted

In addition, 73% of parents in Cluster 1 are unvaccinated, while 61% and 59% of parents in Clusters 2 and 3 are unvaccinated, respectively. However, we find that the three types of vaccinated parents (partially vaccinated, fully vaccinated, fully vaccinated and boosted) are relatively evenly distributed across the three clusters.

## Discussion

In this study, we aimed to identify the determinants of parental COVID-19 vaccine compliance and the main motivating beliefs for reluctance to vaccinate children among unwilling parents. We found parental vaccination status to be a large predictor of parent preference for child vaccination. In addition, we find heterogeneity among parents both in correlates of willingness and reasons for reluctance. For instance, several correlates differed by parental vaccination status. Among the unvaccinated, younger age and lower education were associated with willingness, and among the vaccinated an inverse association was observed. Reasons for refusal to vaccinate children also differed by parental vaccination status with unvaccinated parents less likely to trust government and scientists. Unwilling parents can be divided into three categories based on their reasons for reluctance. To our knowledge, this is the first study to use a cluster analysis on parental vaccine refusal. Additionally, previous research has suggested that the survey is representative of the larger population in weighted demographic categories [25].

Our findings on the correlates of willingness to vaccinate children against COVID-19 are generally consistent with previous studies. Previous research has found vaccinated parents much more likely than unvaccinated parents to be willing to vaccinate their children [8,12–22]. Prior studies have also found likelihood of child vaccination higher among parents of older children, older parents, higher educated parents, Democrat-affiliated parents, insured parents, higher income parents, Hispanic and Asian parents, parents with routine influenza vaccine behavior, and male gender [12,13,15,21,22,31,32]. Unlike past research, we find lower income parents have higher odds for willingness on three of four multivariate models. We additionally find an inverse relationship between willingness and higher age and education among unvaccinated parents.

Prior research on barriers to COVID-19 vaccination parental compliance have found top reasons are often trust related, including vaccine safety and effectiveness, side effects, and lack of trust in government [12–15]. We find consistent results with past studies in the prevalence of concerns over vaccine safety, side effects, and lack of trust in government. However, we find that for a majority of parents (Cluster 2), side effects rarely factor into the decision to vaccinate children, suggesting non-trust related considerations such as accessibility or apathy as the main factors driving hesitancy among this group. Simply looking at the top concerns across parents then, will ignore important distinctions. Moreover, parents in Cluster 1 and Cluster 3 both have severe trust concerns, with large portions in both groups concerned about side effects and the speed at which the vaccine was developed. Yet a large majority of Cluster 1 parents exhibit a lack of trust in government or scientists while a small minority of Cluster 3 parents exhibit a lack of trust in both. Treating parents with side effect concerns as a homogenous group may ignore important differences and may fail to sufficiently ease their concerns.

Several limitations must be considered in interpreting the study results. First, bias is expected from any observational study conducted using a convenience sample; those willing to participate in our study may not be representative of the public as a whole and our findings may not be generalizable to the broader US population. The use of survey weights and the anonymity of the survey may help lessen this bias and prior work suggests the survey is well-targeted at representing the demographics of the US [25]. However, some non-targeted groups are likely underrepresented such as those lacking internet access. Second, we did not have direct measures of certain variables of interest such as child COVID-19 vaccination status and instead must rely on self-report. However, previous research on this survey's estimates have shown it to track closely with CDC estimates for overall vaccination rates as well as alternative estimates for pandemic behavior such as mask wearing across the US [24,33]. Third, results may be limited by the dynamic nature of the COVID-19 pandemic with changes in public

policy, public health recommendations, and case prevalence during the study period. For instance, parents today may express different preferences towards vaccinations as newer options become more available such as Novavax and new variants circulate. Fourth, this analysis solely focuses on understanding vaccine preferences at the individual-scale and ignores the dynamics of community- and region-level factors. Prior research highlights the importance of studying these dynamics at the community level with spatial clustering potentially leading to outbreaks even when aggregate vaccination coverage is high [34].

Willingness to vaccinate children is associated with many different parental characteristics with parent vaccination status being a major determinant. The results presented here can inform the further work needed to tailor child vaccination efforts to subgroups of parents who are more reluctant. It is important to acknowledge that predictors of parental vaccine willingness differ among vaccinated and unvaccinated parents and that examining unwilling parents' concerns in aggregate will obscure many important distinctions.

## Supporting information

**S1 Fig. Subject selection flowchart.**
(EPS)

**S1 Table. Multivariate results for parents with children under 5 and no children under 5.**
(DOCX)

**S2 Table. Reasons for refusal by cluster.**
(DOCX)

**S3 Table. Characteristics of parents with uncertain or inconsistent preferences.**
(DOCX)

**S4 Table. Percent willing to vaccinate children.**
(DOCX)

**S5 Table. Multivariate relationship between parent characteristics and willingness to vaccinate children, varying reference level for parent vaccination status.**
(DOCX)

**S6 Table. Multivariate results, stratification by age.**
(DOCX)

**S7 Table. Multivariate results, stratification by education.**
(DOCX)

**S8 Table. Multivariate results, stratification by income.**
(DOCX)

**S9 Table. Multivariate results, stratification by political party.**
(DOCX)

**S10 Table. Reasons for refusal by child age.**
(DOCX)

## Acknowledgments

The authors thank Kara Sewalk and the SurveyMonkey research team (Laura Wronski, Tim Gravelle & Jon Cohen) for their assistance. Mr. Sehgal had full access to all the data in the study and takes responsibility for the integrity of the data and the accuracy of the data analysis.

## Author Contributions

**Conceptualization:** Neil K. R. Sehgal, Benjamin Rader, Christina M. Astley, John S. Brownstein.

**Data curation:** Neil K. R. Sehgal, Benjamin Rader, John S. Brownstein.

**Formal analysis:** Neil K. R. Sehgal, Benjamin Rader.

**Funding acquisition:** Benjamin Rader, Autumn Gertz, Christina M. Astley, John S. Brownstein.

**Investigation:** Neil K. R. Sehgal, Benjamin Rader, Autumn Gertz.

**Methodology:** Neil K. R. Sehgal, Benjamin Rader.

**Project administration:** Benjamin Rader, Autumn Gertz, John S. Brownstein.

**Resources:** Autumn Gertz, Christina M. Astley, John S. Brownstein.

**Software:** Neil K. R. Sehgal, Benjamin Rader.

**Supervision:** Benjamin Rader, Christina M. Astley, John S. Brownstein.

**Validation:** Neil K. R. Sehgal, Benjamin Rader.

**Visualization:** Neil K. R. Sehgal, Benjamin Rader.

**Writing – original draft:** Neil K. R. Sehgal.

**Writing – review & editing:** Neil K. R. Sehgal, Benjamin Rader, Autumn Gertz, Christina M. Astley, John S. Brownstein.

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
