## [Decision Letter · Decision Letter 0]

12 Dec 2022

PDIG-D-22-00306

Parental Preferences and Reasons for COVID-19 Vaccination Among Their Children

PLOS Digital Health

Dear Dr. Sehgal,

Thank you for submitting your manuscript to PLOS Digital Health. After careful consideration, we feel that it has merit but does not fully meet PLOS Digital Health's publication criteria as it currently stands. Therefore, we invite you to submit a revised version of the manuscript that addresses the points raised during the review process.

Please, pay particular attention to the comments by Referee #3 and #4, who raised some relevant concerns. 

Please submit your revised manuscript within 60 days Feb 10 2023 11:59PM. If you will need more time than this to complete your revisions, please reply to this message or contact the journal office at digitalhealth@plos.org. Please include the following items when submitting your revised manuscript:

We look forward to receiving your revised manuscript.

Kind regards,

Michele Tizzoni

Academic Editor

PLOS Digital Health

Journal Requirements:

1. We ask that a manuscript source file is provided at Revision. Please upload your manuscript file as a .doc, .docx, .rtf or .tex.

2. Please provide separate figure files in .tif or .eps format only and remove any figures embedded in your manuscript file. Please also ensure that all files are under our size limit of 10MB.

Additional Editor Comments (if provided):

Reviewers' comments:

Reviewer's Responses to Questions

**Comments to the Author**

1. Does this manuscript meet PLOS Digital Health’s publication criteria? Is the manuscript technically sound, and do the data support the conclusions? The manuscript must describe methodologically and ethically rigorous research with conclusions that are appropriately drawn based on the data presented.

Reviewer #1: Yes

Reviewer #2: Yes

Reviewer #3: Partly

Reviewer #4: Partly

2. Has the statistical analysis been performed appropriately and rigorously?

Reviewer #1: I don't know

Reviewer #2: Yes

Reviewer #3: Yes

Reviewer #4: No

3. Have the authors made all data underlying the findings in their manuscript fully available (please refer to the Data Availability Statement at the start of the manuscript PDF file)?

Reviewer #1: Yes

Reviewer #2: Yes

Reviewer #3: Yes

Reviewer #4: Yes

4. Is the manuscript presented in an intelligible fashion and written in standard English?

Reviewer #1: Yes

Reviewer #2: Yes

Reviewer #3: Yes

Reviewer #4: Yes

5. Review Comments to the Author

Reviewer #1: This paper aim to characterize factors of parental vaccination hesitancy among their children using data from large-scale US surveys. They found that the willingness to vaccinate a child for COVID-19 was strongly associated with parental vaccination status. In particular, parental age and education is positively associated with willingness for vaccinated parents. Last, they characterized three groups of hesitant parents based on their reasons for vaccinate hesitancy. empirical data provides insights on the role of complex social processes in driving spatial heterogeneity in vaccine hesitancy. This observational study provides interesting insights on factors driving heterogeneity in vaccine hesitancy, but, at this stage, it cannot be published.

I have a few recommendations for improvement or clarification.

Major Comments

1) In the paper, you discussed as in previous studies the sample of survey participants were corrected to reflect the US adult population in demographic composition and political beliefs. After the filtering, how did you re-weight the sample? I think this sentence “After exclusion, the mean survey weight of participants was .98, leading to a weighted sample of 29,583“ needs clarifications, and the manuscript would benefit of a panel with figures showing statistics on respondents before and after filtering.

2) Di you test cross-correlation among indicators used in the analysis, such as between “less education” and “a lack of health insurance”? I expected these are highly correlated, and I think the manuscript will benefit about more insight on the interconnection of some factors. 

Minor Comments

1) Please add in this sentence of the abstract ‘Among hesitant parents, parental vaccination status was inversely associated with reported lack of trust in government (p<.001) and scientists (p<.001)‘, the values of the negative significant correlation.

2) please take in consideration to cite this work on Spatial clustering in vaccination hesitancy: https://journals.plos.org/ploscompbiol/article?id=10.1371/journal.pcbi.1010437

Reviewer #2: This study explores the reasons for vaccine hesitancy in parents regarding Covid-19 vaccination among their children. 

This work relies on data from cross-sectional surveys representing a robust sample of the US population. The paper is well-written and easy to follow and the statistical analysis is technically sound. 

I have a few suggestions regarding the method's section: 

1. I would recommend writing the equation for the multivariate logistic regression as log(P(y=1)/P(y=0))=\\alpha+\\beta... rather than Y=....

2. Can the authors provide more details about the weights W and how are they implemented?

3. I appreciate that the authors shared the data to perform the analysis, and it would be useful to provide the code on github too.

4. Third paragraph in the Discussion section, line 2: oft should be often?

Reviewer #3: The study and manuscript examine reasons related to parent/caregiver COVID-19 vaccination for their children. Data from cross-sectional surveys undertaken between January 1 and May 9, 2022, were pooled to create a sample of 30,174 U.S. parents with at least one child of COVID-19 vaccine eligible age (5-17 years old). Respondents answered questions about demographics, pandemic-related behavior, and COVID-19 self and child vaccination status. Major findings included: fully vaccinated and unvaccinated parents reported concordant vaccination preferences for their eligible child; high parent age and education were positively associated with willingness to vaccinate their children; and cluster analysis revealed three groups of hesitant parents.

Comments and suggestions to the authors:

* The use of the phrase "parental preferences" in the title is confusing given the focus of this study was on identifying factors, primarily demographic, related to parents' compliance with COVID-19 vaccination recommendations for their 5 to 17 year-old children. The word "preferences" does not appear in the Introduction or Survey Question areas of the manuscript. It would be better to frame this as an effort to identify factors associated with parent compliance.

* While the Abstract and Introduction mention the word hesitancy, the term is never defined. The lack of a definition is particularly problematic given a) it does not appear that any survey questions sought to measure hesitancy (e.g., parent doubt or reluctance regarding COVID-19 vaccines or vaccination) and b) the analyses did not include parents who indicated their were "unsure" about having their child receive a COVID-19 vaccine/vaccination. It seems that parents who are "unsure" best reflect actual hesitancy (i.e., they are currently reluctant), whereas parents who have declined or do not intend to have their children receive a COVID-19 are more accurately characterized as "decliners" or "refusers." It is pretty clear these parents have made a clear decision with respect to vaccination and thus are not actually hesitant. This research and manuscript would be much more helpful if it accurately characterized parents and included "unsure" parents in the analyses.

* It is worrisome that in the Introduction the authors state that prior research has been "limited by potentially non-representative samples," when, as indicated in the Discussion, one of the limitations of this study is that uses a "convenience sample" . . . "that may not be representative of the public as a whole." In addition, while the authors state in the Discussion that "previous research has suggested that the survey is representative of the larger population [25]," the study cited (i.e., 25) a) states upfront that the data collection method is "an ongoing prospective, nonprobability-based cross-sectional online survey" (which is a better characterization) and b) mentions as a first limitation "the survey uses a nonprobability-based sample and the results might not be generalizable to the U.S. population."

* The relatively short Discussion devotes an entire paragraph to the WHO 3 Cs framework, but that framework appears to be applied post hoc (i.e., it was not the basis for survey questions or measures). At a minimum, if the 3 Cs framework is to be used it should have been introduced in the Introduction and set up in the Methods section. 

* It is stated in the Discussion that the results suggest "better tailoring" of messages to increase vaccination rates among children but no specifics or examples are provided. As such, this guidance has almost no value.

* While the tables provide much information, the information in the tables is difficult to interpret. It would be helpful if Table 2 included the percentages as well as the odds ratios and indicated which were statistically significant (e.g., via bolding). Tables 3 does not indicate which results are statistically significant within clusters or between clusters. The same issues are true for the appendix tables.

Reviewer #4: In this paper, dr Sehgal and colleagues investigate the willingness to vaccinate children from a survey of US self-reported parents. They found that parent vaccination, as well as a many other parental characteristics, is associated with an increase willingness to vaccinate their children. I would be grateful if the authors could address my questions and comments regarding their approach.

I suggest author to not define parent vaccination as a determinant for the willingness, more likely, the set of characteristics that increase the likelihood of being vaccinated in parents also increase the likelihood of being willing to vaccinate the children. I would reframe the discussion and the analysis to better investigate/highlight this aspect

Given the broad range of epidemiological condition worldwide, I would suggest authors to make explicit both in the title and in the background that the data are from only one country, the United States. Moreover, I would suggest authors to better frame their findings with respect to vaccination coverage for children in other countries, is it the lower rate only a US problem?

Method: “Each day, over 2 million individuals complete online surveys” where? In the US? Was the survey anonymous?

Survey Questions. 

how have authors addressed parents who already have vaccinated their children? Likewise, why restricting the analysis to parents always willing or unwilling to vaccinate all of their children? How many were they? Would it possible to analyse the reasons of such differential behavior (maybe some side-effects after vaccinating first child? Increase exposure to side-effects news? Maybe let older children decide for themselves)?

given that authors restricted the analysis the following sentence should be “Y is a binary variable for the willingness of parent I to vaccinate” ALL “their child(ren)”? Regarding Y, how the “not sure” answer were treated?

X is equal to 1 if parent is vaccinated, please define here what vaccinated is, eg. At least one dose, two, booster.. also, define all terms that are written in the equation (eg, alpha, beta, gamma..)

Could it possible that only partially vaccinated parents, despite being classified as vaccinated have developed opposing view with respect to vaccination and therefore be less willing to vaccinate their children? Would it be possible to explore such hypothesis?

Equation 1, why authors did not consider any interaction effects between the vaccination status and other parent level factors? In the present form and if I understand correctly, the equation assumes that the associated effect (beta) of being vaccinated to the willingness to vaccinate is constant across any other parental difference (eg, authors are not testing if the increase in willingness to vaccinate associate with vaccinated parents is different between high income and low income parent). It seems to me that this is a strong modelling assumption that should discussed and tested. 

Did authors consider if any collinearity between variables (eg higher educated or higher income parents more likely to be also vaccinated) was present? Could such collinearity bias the association with the response?

R version, please include citation, code availability statement and package used if any

Is an ethic committee required for such type of studies? is the Boston Children’s Hospital Institutional Review Board an ethics committee qualified for approval? Shouldn’t be included some kind of reference/document number of the approval? 

Result

I would like to have a table reporting the actual response on willingness to vaccinate, not only the odds ratio. Contrast are given considering unvaccinated as reference, but multiple comparison may be interesting, eg is there a statistical difference between fully and partially vaccinated parents?

Discussion

If authors want to give an idea of the representativeness of their sample and generalizability of their study they may want to train their regression on a 80%subset of their data and then test the predictive performance on the other.

6. PLOS authors have the option to publish the peer review history of their article (what does this mean?). If published, this will include your full peer review and any attached files.

**Do you want your identity to be public for this peer review?** For information about this choice, including consent withdrawal, please see our Privacy Policy.

Reviewer #1: Yes: Giulia Pullano

Reviewer #2: No

Reviewer #3: No

Reviewer #4: No

---

## [Decision Letter · Decision Letter 1]

14 Mar 2023

Parental Compliance and Reasons for COVID-19 Vaccination Among American Children

PDIG-D-22-00306R1

Dear Mr. Sehgal,

We are pleased to inform you that your manuscript 'Parental Compliance and Reasons for COVID-19 Vaccination Among American Children' has been provisionally accepted for publication in PLOS Digital Health.

Best regards,

Michele Tizzoni

Academic Editor

PLOS Digital Health

Reviewer Comments (if any, and for reference):

Reviewer's Responses to Questions

**Comments to the Author**

1. If the authors have adequately addressed your comments raised in a previous round of review and you feel that this manuscript is now acceptable for publication, you may indicate that here to bypass the “Comments to the Author” section, enter your conflict of interest statement in the “Confidential to Editor” section, and submit your "Accept" recommendation.

Reviewer #1: All comments have been addressed

Reviewer #2: All comments have been addressed

Reviewer #3: All comments have been addressed

Reviewer #4: All comments have been addressed

2. Does this manuscript meet PLOS Digital Health’s publication criteria? Is the manuscript technically sound, and do the data support the conclusions? The manuscript must describe methodologically and ethically rigorous research with conclusions that are appropriately drawn based on the data presented.

Reviewer #1: Yes

Reviewer #2: Yes

Reviewer #3: Yes

Reviewer #4: Yes

3. Has the statistical analysis been performed appropriately and rigorously?

Reviewer #1: Yes

Reviewer #2: Yes

Reviewer #3: Yes

Reviewer #4: Yes

4. Have the authors made all data underlying the findings in their manuscript fully available (please refer to the Data Availability Statement at the start of the manuscript PDF file)?

Reviewer #1: Yes

Reviewer #2: Yes

Reviewer #3: Yes

Reviewer #4: Yes

5. Is the manuscript presented in an intelligible fashion and written in standard English?

Reviewer #1: Yes

Reviewer #2: Yes

Reviewer #3: Yes

Reviewer #4: Yes

6. Review Comments to the Author

Reviewer #1: Authors are largely answered and clarified all comments in my review. The paper is easier to follow and methods are fully explained.

Reviewer #2: The authors addressed all of my comments and made the code and data available. Minor point: the equation is in a weird format, please make sure the format is readable.

Reviewer #3: The authors have done an excellent job of addressing the reviewers' comments and suggestions. This is a well structured and written manuscript, and the findings should be of much value to those involved in efforts to foster parent compliance with vaccination recommendations for children.

Reviewer #4: Authors addressed all my previous comments.

In the revised manuscript they reformulated the equation as, I think, suggested by another reviewers, however it is now only partially visible (see Line 119 equation formula)

7. PLOS authors have the option to publish the peer review history of their article (what does this mean?). If published, this will include your full peer review and any attached files.

**Do you want your identity to be public for this peer review?** For information about this choice, including consent withdrawal, please see our Privacy Policy.

Reviewer #1: No

Reviewer #2: No

Reviewer #3: No

Reviewer #4: No
